# Silymarin’s Inhibition and Treatment Effects for Alzheimer’s Disease

**DOI:** 10.3390/molecules24091748

**Published:** 2019-05-06

**Authors:** Hong Guo, Hui Cao, Xiaowei Cui, Wenxiu Zheng, Shanshan Wang, Jiyang Yu, Zhi Chen

**Affiliations:** School of Pharmacy, Shandong University of Traditional Chinese Medicine, Jinan 250355, China; 13589080261@163.com (H.G.); 13553158985@163.com (H.C.); 13553159162@163.com (X.C.); m17862987280@163.com (W.Z.); 17862986901@163.com (S.W.); 15621875760@163.com (J.Y.)

**Keywords:** silymarin, Alzheimer’s disease (AD), amyloid β-protein (Aβ), acetylcholine (ACh), oxidative stress, neuroinflammatory

## Abstract

As a longstanding problem, Alzheimer’s disease (AD) has stymied researchers in the medical field with its increasing incidence and enormous treatment difficulty. Silymarin has always been valued by researchers for its good efficacy and safety in treating liver disease. Recent studies have shown that silymarin also has good pharmacological activity in the nervous system, especially for the treatment of AD. Silymarin can control the production of Aβ by inhibiting the precursor substance of Aβ (β-amyloid precursor protein), and it can inhibit the polymerization of Aβ. Silymarin can also increase the acetylcholine content in the nervous system by inhibiting cholinesterase activity. At the same time, it also has the effect of resisting oxidative stress and the inflammatory response of the nervous system. These pharmacological activities contribute to the inhibition of the onset of AD. The good efficacy of silymarin on AD and its high safety and availability give it huge potential for the treatment of AD.

## 1. Introduction

Alzheimer’s disease (AD) is a neurodegenerative disease that was first reported in 1907 by the German psychiatrist Alois Alzheimer, of whom it is named after. Over the past century, the incidence of AD has been rising, but unfortunately, so far, we have not found a complete cure for AD [1]. At present, it has become one of the most serious diseases that threatens the health and safety of the elderly; besides cardiovascular diseases, cerebrovascular diseases, and tumors. Patients with AD have cognitive and memory dysfunction, personality, behavioral and other performance issues. Patients usually die within about ten years.

The specific pathogenesis of AD is not yet fully understood, but there are many factors that can influence it and the mechanism of action is complex (Figure 1). Physiologically, the most important feature in AD patients is the loss of a large number of neurons in the brain and the destruction of the neuronal synaptic function. This symptom is mainly found in the hippocampus, noting that the hippocampus is also a vital area in the process of memory formation [2]. The two most typical pathological changes in AD are: amyloid β-protein (Aβ) abnormal deposition of senile plaques (SPs) and tau protein abnormally phosphorylated (tau-ap) to form neurofibrillary tangles (NFTs) [3]. Besides, neuroinflammation, abnormal immune regulation, oxidative stress, gene, calcium ion, central cholinergic system dysfunction and insulin signaling pathway disorders are also closely related to the pathogenesis of AD. Among them, Aβ is usually used as the basis for diagnosing AD. Its position in the pathogenesis of AD is extremely important. Based on this amyloid hypothesis we believe that Aβ is the causative agent of AD, this hypothesis is also accepted by most people [2]. Current research indicates that the oligomeric fibril precursors of Aβ are the main substances that cause pathological changes [4]. The ability to destroy cell membranes of Aβ is an important mechanism of its neurotoxicity. There are different opinions about the mechanism by which it destroys cell membranes. That is, its destruction of the cell membrane may be achieved in two steps: First, ion-selective pores are formed, and this is followed by non-specific cleavage of the lipid membrane during the formation of amyloid fibers [5,6].

Recent related studies have found that *Porphyromonas gingivalis* is also associated with the onset of AD [7,8]. The gingipains produced by *Porphyromonas gingivalis* are a toxic protease that causes the production of Aβ1-42, which is toxic to tau. At the same time, the infection of the bacteria causes neuroinflammation in the brain. There are also studies showing that the dynamic balance of metal ions is closely related to the pathogenesis of AD, especially Zn2+ and Cu2+. These two metal ions participate in the declarative memory and spatial memory process in the hippocampus. It has been found that these metal ions have an elevated concentration in the brain of AD patients, especially related to the accumulation of Aβ [9]. These ions have a defined binding site with Aβ, where they bind to Aβ and effect its polymerization and toxicity, which in turn effects the progression of AD [10,11,12,13]. 

*Silybum Marianum* (L.) Gaertn is a herbaceous plant of the Campanulales, Compositae, Silybum Adans. It is native to Europe, the Mediterranean, North Africa, and Central Asia. The earliest record of *Silybum Marianum* (L.) Gaertn can be traced back to Europe in the first century. Silymarin is a mixture of flavonoids extracted from the seeds of *Silybum Marianum* (L.) Gaertn. It mainly includes silybin A, silybin B, isosilybin A, isosilybin B, silydianin, silychristin, and taxifolin (Figure 2). Silybin is one of the most abundant and active ingredients in silymarin. Ever since it was discovered by German pharmacologist H. Wagner et al. in the 1960s, it has been a research hotspot because of its excellent liver protection ability. In addition, it also has functions in lowering blood lipids, lowering blood glucose, anti-oxidation, anti-tumor, anti-inflammation, and anti-virus [14]. At the same time, according to relevant studies in recent years, silymarin also has good effects in the fight against AD.

## 2. Aβ Inhibitor

### 2.1. Aβ and the Onset of AD

Abnormal deposition of β-protein (Aβ) is one of the most central lesions in the pathogenesis of AD. At present, the Aβ cascade reaction theory is the most recognized theory in the various hypotheses of AD pathogenesis. 

Aβ is a short polypeptide with a molecular mass of approximately 4 k Da, with approximately 39–43 amino acid units. In vivo, Aβ is produced by the sequential hydrolysis of β-amyloid precursor protein (APP) by β-secretase and γ-secretase and it can be removed by hydrolysis of the Aβ enzyme. APP is an important substance that has a key role in the nervous system. It can plasticize nerve cells and regulate the ability of nerve cells to connect. It is also associated with cell growth and adhesion. Therefore, APP is ubiquitous in a variety of cells, especially in nerve cells, and it is continuously expressed and produced [15]. When these cells undergo normal metabolism, it is easy to convert APP into Aβ. Aβ (comprising of Aβ40 and Aβ42, where Aβ42 is more prone to amyloidosis and Aβ40 is more likely to form oligomers of Aβ) has complex toxic mechanisms for neuronal cells [16]. The formation and degradation of Aβ is in a dynamic equilibrium, so that the content of Aβ is relatively stable. When this balance is broken, the production of Aβ is excessive and the decomposition ability is insufficient, which causes a large amount of Aβ accumulation and sedimentation, thereby forming amyloid plaques (also known as SPs) and an Aβ oligomer. At present, the pathogenesis of AD can be mainly divided into two types: family type and sporadic type. Family type AD is mainly related to genes, where the coding genes of APP and γ-secretase (mainly the active central protein presenilin) mutation cause the excessive production and accumulation of Aβ, which leads to the onset of AD. The sporadic AD, in addition to the excessive production of Aβ, also decreases the decomposition efficiency of Aβ, which causes the accumulation of Aβ, leading to the onset of the disease [17]. The former is relatively early onset and the latter is late, wherein the latter is also caused by the decline in Aβ clearance ability after aging. This is one of the reasons why AD is more common in the elderly population. With recent studies, the role of SPs in the pathogenesis of AD may not be as important as previously thought, and the Aβ oligomers produced by cellular metabolism or Aβ aggregation may have greater harm [18]. This oligomer, that is highly toxic to the nervous system, primarily impairs the patient’s ability to learn and remember through damaging the synapse. It can cause great damage to the number of synapses in the neuronal cells and the number of synaptic receptors, which ultimately leads to nerve connection damage and reduces synaptic transmission [19,20,21,22,23]. This ability for synaptic damage is complicated and is largely related to four signaling pathways: N-methyl-D-aspartic acid (NMDA) receptor pathway; metabotropic glutamate R5 (m Glu R5) pathway; Eph/Ephrin pathway and the c AMP-response element (CRE) pathway. At the same time, the oligomer of Aβ can also cause the tau protein to be abnormally phosphorylated [18], resulting in further damage to the nervous system. Tau protein is a microtubule-associated protein that is mainly localized to neuronal axons under normal conditions. Its main function is to promote the formation of microtubules and maintain the stability of microtubules. Abnormal phosphorylation of tau protein affects its ability to form microtubules, while NFTs are induced to induce AD [24]. In addition, Aβ also induces a large amount of inducible nitric oxide synthase in colloidal stellate cells by caspase. Eventually it produces a large amount of NO that damages the nerve cells. At the same time, Aβ has a strong destructive effect on the cell membrane, wherein it can insert into the cell membrane and destroy the homeostasis and integrity of the cell membrane, eventually leading to apoptosis of the nerve cells [25]. Recent studies have shown that this cell membrane damage caused by Aβ can be inhibited by Ca ions. However, the Ca ion is a double-edged sword, which also causes an increase in membrane rupture associated with fibers grown on the surface of the lipid membranes [26].

In addition to the above hazards, Aβ can also damage the nervous system by mediating oxidative damage [27], neuroinflammation [28,29,30], and neuronal apoptosis [31] to cause AD production.

Reducing the production of Aβ and promoting the degradation of Aβ is one of the most important actions in delaying the onset of AD and treating AD.

### 2.2. Inhibition of Silymarin on Aβ and Aβ Aggregate Products

Silymarin is an Aβ inhibitor [32], especially silybin [33]. In vitro experiments have demonstrated that silymarin can effectively inhibit the formation of Aβ fibrils and the toxicity of Aβ to nerve cells. Murata N et al. found in their study of APP transgenic mice, that silymarin could significantly reduce the deposition of Aβ in the brain and it could effectively improve the behavioral abnormality of mice. At the same time, silymarin also significantly reduced the content of Aβ oligomers in the nervous system [32]. Yin F et al. found that silybin could inhibit the polymerization of Aβ in a dose-dependent manner, and that silybin could inhibit the neuronal damage caused by Aβ-induced oxidative stress [33].

The overexpression of APP is closely related to the pathological changes of AD [34,35]. Excess APP as a precursor of Aβ production will promote the production of Aβ, leading to the accumulation and self-assembly of Aβ. This phenomenon is largely related to gene presentation, which is also an important cause of Aβ accumulation in the pathogenesis of familial AD. Moreover, studies have shown that APP expression shows an upward trend in the aging brain [36]. This in turn represents the excessive production of APP, which is one of the key factors in the Aβ accumulation process of sporadic AD. Since APP is a precursor substance for the production of Aβ, the control of APP expression can be used as an effective method to regulate the amount of Aβ production. In the process of searching for AD drugs, the regulation of APP production and the inhibition of gene overexpression are important ideas. 

Silymarin can effectively inhibit the expression of the APP gene, and it also has the ability to disintegrate already formed Aβ plaques [37] (Figure 3). Yaghmaei P et al. directly injected Aβ into the bodies of Wistar rats to create an animal model of AD. After treatment with silymarin, behavioral experiments with the rats showed that the AD status had significantly improved. At the same time, a comparison between the model group and the animals in the drug-administered group showed that the Aβ plaques in the brain of the silymarin treated rats were significantly reduced, and the expression of the APP gene was also inhibited [37]. This indicated that silymarin could improve the condition of AD by disintegrating the already produced Aβ plaque, and it could also block the synthesis of Aβ by reducing the APP content in the nervous system, through inhibiting the expression of the APP gene and reducing the accumulation of Aβ from the source. The inhibitory effect of silymarin on Aβ was simultaneously carried out by controlling both its production and accelerating its decomposition, which is bound to have better therapeutic effects and speed.

Silymarin also has an inhibitory effect on the self-assembly of Aβ to avoid the conversion of accumulated Aβ into more toxic SPs and oligomers. As a mixed substance, different components of silymarin have different effects on the self-assembly of Aβ. This is closely related to the unique structure of each component. Additionally, 3′,4′-dihydroxy in the silymarin structure has a good inhibitory effect on the Aβ self-assembly process. The 7-position hydroxyl group and the 2,3-position stereochemical structure are relatively less important in inhibiting the self-assembly of Aβ [38].

Amongst the various components of silymarin, silybin is the best and most abundant. There are two stereoisomers in natural silybin: silybin A and silybin B. Although they have a good effect in inhibiting the formation of Aβ mature fibrils, the nuances of their stereochemical structures give them different mechanisms of action. Sciacca MFM et al. studied the effects of silybin’s stereochemistry on its inhibitory effect on Aβ and found that silybin A preferentially binds to the ^27^ NKGAII ^32^ and ^17^ LVFF ^20^ sequences of Aβ40, while silybin B binds mainly to the ^35^ MVGGVV ^40^ sequence of Aβ40. At the same time, 2,3-dehydrosilybin A and 2,3-dehydrosilybin B are preferentially combined with sequence ^17^ LVFF ^20^. Sciacca MFM et al. also found that silybin B is the only one of these four components to inhibit the formation of Aβ through inhibiting the growth of small-sized protofibrils. Silybin is also the most effective inhibitor of Aβ toxicity, it can be concluded that stereochemistry is crucial in the process of silymarin inhibiting Aβ [39].

## 3. Enhancing Cholinergic Energy

### 3.1. ACh Deficiency and the Onset of AD

Another hypothesis for the pathogenesis of AD, which is recognized by many scholars, is the cholinergic hypothesis. It was also the earliest hypothesis on the pathogenesis of AD. This hypothesis suggests that the onset of AD begins with the lack of the neurotransmitter acetylcholine (ACh) (Figure 4). Neurotransmitters in the nervous system are essential substances that ensure the transmission of neural signals between different neurons. Many neurotransmitters are directly involved in the construction and maintenance of learning and memory in the brain. Amongst these neurotransmitters, ACh is the most classic one, and it is also the neurotransmitter that has the greatest influence on learning and memory. The cholinergic receptor system is one of the most important nerve conduction pathways in the human body. It plays an important role in learning and memory when combined with acetylcholine [40,41]. Under normal circumstances, the synthesis and decomposition of ACh is in a dynamic balance to ensure the normal operation of the nervous system. Imbalances in the synthesis and decomposition of ACh in the nervous system of AD patients cause a relative lack of ACh content, which impairs learning and memory. ACh in the brains of AD patients is in a relatively deficient state, which renders significant damage to their learning and memory ability. As the aging process progresses, some neurons in the brain become increasingly difficult to regenerate, and the choline content decreases as well, which directly leads to a decrease in the synthesis of ACh. In the case of a decrease in the amount of ACh synthesis, cholinesterase accelerates the breakdown of ACh, which in turn causes a rapid decrease in ACh content in the brain.

Based on this theory, increasing the content of ACh in the nervous system could greatly alleviate the condition of AD. In general, the promotion or inhibition of intracellular metabolites is mostly achieved through increasing the amount of their production and inhibiting their decomposition. In addition, increasing the amount and activity of ACh receptors is also an effective method. In contrast, there are many factors influencing the pathway of ACh synthesis [42,43,44], and the pathway to reduce the decomposition of ACh through acetylcholinesterase (AChE) inhibitors is considered more direct and effective. 

Interestingly, AChE inhibitors also have an inhibitory effect on the aggregation and toxicity of Aβ [45,46]. This may be because AChE has the effect of inducing Aβ aggregation [47,48]. AChE-induced Aβ polymerization is achieved via its peripheral anion site, which combines with Aβ through electrostatic attraction to form a highly toxic acetylcholinesterase–Aβ peptide [49,50]. Then it induces the transformation of Aβ conformation to β-sheet to promote fibril formation. In addition, AChE can also cause and promote the occurrence of neuroinflammation [47]. Consequently, AChE inhibitors can alleviate AD in a variety of ways.

### 3.2. Silymarin Inhibits AChE and BChE

At present, AChE and BChE (BChE is also known as Pseudocholinesterase, PChE, which has a destructive effect on ACh, and it has been found that its activity increases with age and the development of AD [51]) inhibitors have been used in the clinical management of AD and have achieved good results [52]. The vast majority of drugs that have been used in the clinical treatment of AD are AChE inhibitors. For example, Metrifonate, Donepezil, Galantamine, Tacrine, and other first line drugs in the treatment of AD are AChE inhibitors [41,45].

Silymarin has a good inhibitory effect on AChE and BChE [52,53,54]. It can increase the ACh content by inhibiting the activity of AChE and BChE, thereby alleviating the AD condition.

Using scopolamine (Figure 4) to induce experimental and memory impairment in experimental animals is a classic animal model in AD research. It is an animal model for AD research based on the cholinergic hypothesis. This is because scopolamine is a type M acetylcholine receptor and a competitive antagonist. By injecting scopolamine, it can compete with acetylcholine to occupy the acetylcholine receptors and mimic the state of acetylcholine deficiency in the nervous system, leading to learning and memory impairment. *Silybum marianum* (L.) Gaertn’s methanol extract [53] and silymarin [52] have good therapeutic effects on scopolamine-induced AD animal models. Cholinergic dysfunction in dementia mice induced by scopolamine is mainly manifested by increased activity of AchE [55]. After treatment with silymarin, the activity of AChE in the nervous system was significantly reduced, and cholinergic activity was increased. At the same time, in vitro, the methanol extract of *Silybum marianum* (L.) Gaertn [53] and silymarin [54] also showed significant inhibition of AChE activity. El-Marasy SA et al. used silymarin, to pretreat mice, made from scopolamine and found that it could effectively reduce the activity of AChE, and effectively alleviate the memory damage caused by scopolamine over a short period [55]. Its anti-AD effect is similar to the clinical application of the anti-AD drug Donepezil.

In addition, silymarin also has a good inhibitory effect on BChE [52], where Ilkay Orhan et al. found that silybin inhibited BChE by 51.4%.

## 4. Oxidative Stress Protectant

### 4.1. Oxidative Stress and the Onset of AD

Oxidative stress is an imbalance between oxidative and antioxidant capacity in the body. The aging of the body and most aging-related diseases (such as AD) are related to oxidative stress. The body will produce many highly active molecules with oxidative properties, such as ROS (reactive oxygen species) and RNS (reactive nitrogen species). At the same time, there are two major antioxidant systems in the body: enzymatic and non-enzymatic antioxidant defensive systems. Oxidative stress occurs when the oxidation and antioxidant capacity are imbalanced and tend to oxidize. In this state, the body’s antioxidant system cannot promptly remove the oxidizing substances represented by free radicals generated in the body, causing them to accumulate, and consequently attacking and adversely affecting the body. Compared to other tissues of the body, due to the specialty of the nervous system (high oxygen consumption, high content of polyunsaturated fatty acids [56], high levels of free iron ions, and low levels of antioxidant defense [57]), it is more susceptible to oxidative damage. Recent studies have increasingly shown that the occurrence of AD is inseparable from oxidative stress [58,59,60]. 

In the brains of AD patients, glutathione peroxidase (GSH-Px), superoxide dismutase (SOD), and catalase (CAT) were significantly lower than normal, and the degree of lipid peroxidation was significantly increased [61]. Oxidative stress can damage cells by oxidizing cell membrane lipids and proteins in cells [62], and the ROS and RNS produced by oxidative stress can also induce neuronal damage and apoptosis through regulating the expression of caspase protein [63]. In addition to directly damaging the cell structure and causing apoptosis, oxidative stress can also affect normal neuronal function by affecting the glucose transporter on the cell membrane surface [64,65]. The loss of apoptosis and function of the nerve cells themselves is the direct cause of AD. 

Oxidative stress in the brain causes a series of consequences that affect learning and memory, such as: neuroinflammation, glucose dysmetabolism, elevated levels of free calcium, DNA and RNA damage, etc. (Figure 5). Moreover, oxidative stress is cross-linked with the Aβ cascading theory, tau protein abnormally phosphorylated, and the cholinergic hypothesis, etc. (Figure 5). Aβ itself promotes oxidative stress and mediates intracellular ROS production [66]. In Aβ, Aβ25-35 is the most active fragment, which can be used as a lipid peroxidation initiator to induce a large number of free radicals within a few minutes, causing severe oxidative damage. It has also been found that the aggregation of Aβ can be inhibited by pretreatment with an antioxidant [67]. This suggests that we can alleviate Aβ-induced neurological damage by inhibiting oxidative stress. In addition, the destruction of the phospholipid membrane is an important mechanism of Aβ toxicity [25]. Oxidative stress can cause a decrease in the thickness of the lipid bilayers in nerve cells [68]. This lipid bimolecular membrane change affects the process of Aβ insertion into the cell membrane, thereby affecting the destructive effect of Aβ on the cell membrane [25,68,69].

Lipid peroxidation can also cause the production of NFTs. The free radical and calcium ion overload generated during oxidative stress will also accelerate the production of NFTs caused by the abnormal phosphorylation of tau protein [67]. These indicate that many of the pathogenic factors that cause AD may eventually result in damage to the structure and function of the nerve cells through oxidative stress damage.

### 4.2. Silymarin has the Effect of Inhibiting Oxidative Stress and its Induced Nerve Damage

Silymarin has excellent antioxidative stress effects, especially in the nervous system. Many of its pharmacological effects are achieved through antioxidant functions, including its most representative liver-protecting function. Antioxidant ability is an important part in its protection of learning and memory abilities. In the animal model of AD, the neurotoxin manganese (Mn ions) is often used as a substance to induce AD in animals. Oxidative stress is a recognized pathogenic mechanism of Mn ions-induced AD disease. Yassine Chtourou et al. used silymarin to treat rats with Mn ions poisoning and found that silymarin could completely reverse the oxidative stress damage caused by Mn on the rat cerebellum [70]. The enzyme antioxidant defense system (the content and activity of superoxide dismutase, catalase, and glutathione peroxidase) and the non-enzymatic antioxidant defense system (glutathione, total thiols, and vitamin C.) disorder caused by Mn ions poisoning could be reversed by silymarin. This may have been due to the inhibition of silymarin on lipid and protein oxidation in the brain. At the same time, silymarin also has a considerable degree of relief for the pathological changes of Purkinje cells in the rat cerebellum caused by Mn ions.

Silymarin protected the rat brain from oxidative stress by preventing lipid peroxidation and regulating glutathione [71]. Silymarin has excellent anti-lipid peroxidation ability, considering that most of the brain is composed of lipids. Many experiments have shown that silymarin has good inhibition ability for lipid peroxidation induced by CCl_4_, NADPH–Fe^2+^–ADP, tert-butyl hydroperoxide (TBHP), NADPH–Fe^2+^–ADA, and ethanol, amongst others [72]. Silymarin can reduce the free radicals produced by metabolism and scavenge free radicals. At the same time, it is also a lipoxygenase inhibitor. Lipoxygenase is an enzyme that promotes the oxidation of double bonds and participates in the inflammatory response in vivo due to its ability to catalyze arachidonic acid oxidation. This helps it further exert its ability to resist oxidative stress and also shows its anti-inflammatory effect. Silymarin can regulate glutathione (GSH, reduced glutathione) metabolizing enzymes [73], up-regulate GSH levels [74], and conjugation capacity [75]. GSH has good antioxidant capacity and detoxification ability and it has a good inhibitory effect on oxidative stress damage and neurotoxic substances in the nervous system, resulting in the alleviation of impairments to learning and memory caused by oxidative stress. C. Nencini et al. used oral acetaminophen (APAP) to produce a rat brain model of oxidative stress. In this model, the GSH content and activity in the rat brain were significantly decreased, whilst the GSH level was significantly increased in the model treated with silymarin [71]. In addition, in the oxidative stress rat brain model, the activities of antioxidant substances, such as ascorbic acid (AA) and superoxide dismutase (SOD) were significantly reduced. Meanwhile, the activities of oxidizing substances such as malondialdehyde (MDA) and oxidized glutathione (GSSG) were significantly increased. The application of silymarin could effectively inhibit this phenomenon and protect the antioxidant defense system in the body [71].

The oxidation of proteins plays an important role in the early stages of AD [76], noting that the most severe areas of brain damage in AD patients are the cortex and hippocampus. F. Galhardi et al. found that silymarin inhibited the oxidation of proteins in the cerebral cortex and hippocampus of rats. It is interesting to note that when the daily oral dose for rats is 200 mg/Kg, the inhibitory effect on protein oxidation in the cerebral cortex and hippocampus of aged rats is better than that of young rats [77]. This may show that silymarin is more effective in the treatment of senile dementia represented by AD. The mechanism by which silymarin protects the cerebral cortex and hippocampus from oxidative stress has been studied. Recently, Thakare VN et al. found that this may be achieved through the regulation of corticosterone by silymarin [78].

Silymarin is also effective in improving behavioral indicators in dementia mice, whilst improving the specific biochemical indicators of oxidative stress in the brain. Yetunde Onaolapo Adejoke et al. found that acute coadministration of silymarin and aspartame could improve oxidative stress in the mouse brain. At the same time, the Y-maze experiment could also prove its effectiveness in improving the learning and memory ability in mice [79]. This suggests that silymarin could improve learning and memory by improving oxidative stress in the brain.

## 5. Neuroinflammatory Resistance Agent

### 5.1. Neuroinflammatory Response and the Onset of AD

The inflammation cascade plays an important role in the pathogenesis of neurodegenerative diseases, especially AD [80,81]. Previously, neuroinflammation was considered an accessory product caused by other pathogenic causes, but modern research indicates that it is an important cause of AD [82,83]. A large number of inflammatory factors released via the inflammatory reaction, and a series of toxic substances released by excessive activation of astrocytes and microglia due to inflammation, can cause neuronal deformation and apoptosis [84]. Most of the factors that cause AD (Aβ, tau protein abnormally phosphorylated, and oxidative stress) cause damage by inducing nervous system inflammation to damage the nervous system. In addition, neuroinflammation also promotes the development of these pathological changes [85]. 

The inflammatory factors primarily related to AD include IL [86,87,88], TNF-α [89,90], p38 [91], NF-κB [92], etc. These inflammatory factors can aggravate the inflammatory response or accelerate the accumulation of Aβ, promote the phosphorylation of tau protein, and increase the content and activity of AChE. This can further intensify the development of AD and have a damaging effect on nerve cells. Among them, the mechanism of action of IL-1 is the most complicated, and its influence on AD is the largest. In addition to damaging the nervous system, IL-1 can also promote the production and aggregation of Aβ and trigger cholinergic disorders [93]. The inflammatory cells associated with AD are mainly: T cells [94], macrophages [95], astrocytes [96], and microglia [97,98]. Among them, astrocytes and microglia have the greatest influence on neuroinflammation. Most of the time, these inflammatory cells affect the nervous system by secreting inflammatory factors and other toxic substances that are boosted by inflammatory factors. Intracerebral inflammatory factors (such as YKL-40, ICAM-1, VCAM-1, IL-15, etc.) are significantly elevated in various stages of AD onset. Moreover, elevated levels of these inflammatory markers can also increase the risk of AD and accelerate the progression of the disease [99] (Figure 6). Treatment involving inhibiting inflammation of the nervous system has also shown good results in the treatment of AD rats [100].

### 5.2. Silymarin Relieves the Progression of AD by Inhibiting Neuroinflammation

The anti-inflammatory activity of silymarin has long been recognized, and many of its pharmacological activities (such as the ability to protect the liver) are through anti-inflammatory effects [101]. It can slow down and inhibit the progression of neurodegenerative diseases such as AD by inhibiting inflammation of the nervous system. Silymarin can effectively reduce the levels of inflammatory factors in the cerebrospinal fluid (CSF) [102], as well as inhibit IL-6 and TNF-α in the hippocampus [103]. It also reduces the mRNA levels of proinflammatory mediators in the inflammatory response [104,105]. Cellular experiments have also shown that silymarin can inhibit a variety of inflammation-related signaling pathways (such as the NF-κB pathway) [105].

In the brains of AD patients, microglia and astrocytes are also involved in the inflammatory response [85,106,107]. Astrocytes play a major role in the support and nutrition of the nervous system. In recent years, they have been found to be involved in inflammatory reactions in the brain. On the one hand, microglia can phagocytose toxic substances such as Aβ to protect the brain; but on the other hand, they can release inflammatory factors that impair learning and memory. Silymarin and silybin could protect the glial cells from ROS damage, along with the dose-dependent inhibition of glial cell proliferation [108]. For astrocytes, silybin is likely to inhibit astrocyte activation by inhibiting phosphorylation of ERK and JNK to attenuate the neuroinflammatory responses [109]. For LPS-induced microglia activation, silymarin can inhibit NF-kB by inhibiting it and reducing microglia release using inflammatory mediators, such as TNF-α and NO [110].

## 6. Conclusions

The incidence of AD is increasing yearly, and it not only brings pain to patients and their families, but it also results in great losses to society. Natural drug products are showing increasing advantages in AD treatment. *Silybum marianum* (L.) Gaertn has been used in Europe for thousands of years in medicinal history, and the main active ingredients contained in it have also been used clinically to achieve good results, especially in the treatment of liver diseases. Recent studies have shown that silymarin has great potential for the treatment of AD. For the pathogenesis of AD, silymarin can treat AD by inhibiting the formation and development of Aβ, enhancing cholinergic energy, providing oxidative stress protection, and promoting inhibition of neuroinflammation (Figure 7). For tau protein abnormally phosphorylated and insulin signaling pathway disorders, there has not been any relevant research. Silymarin has proven low toxicity and safety, ascertained over several years of research and clinical application. It also has good nervous system related activity, and provides good prospects for the treatment of AD. 

The treatment and protection of the nervous system provided by silymarin could also be applied in the treatment of other diseases, such as Parkinson’s disease (PD). Some causes of PD, such as neuroinflammation [111] and oxidative stress [25], can be inhibited by silymarin. In addition, AD is also closely related to type II diabetes. Studies have shown that patients with AD have a significantly higher risk of developing type II diabetes, and both are likely to have a common pathogenesis [112]. For AD, an insulin signaling pathway disorder is one of the causes of its pathogenesis; whilst Aβ aggregation and oxidative stress are also factors that lead to type II diabetes [8,112,113]. Therefore, the ability of silymarin to treat AD by inhibiting Aβ and oxidative stress may also have good implications on type II diabetes. These uses offer the possibility to apply silymarin for use in treating other diseases.

We hope that the research done in this article will contribute to the research on silymarin and AD.

## Figures and Tables

**Figure 1 molecules-24-01748-f001:**
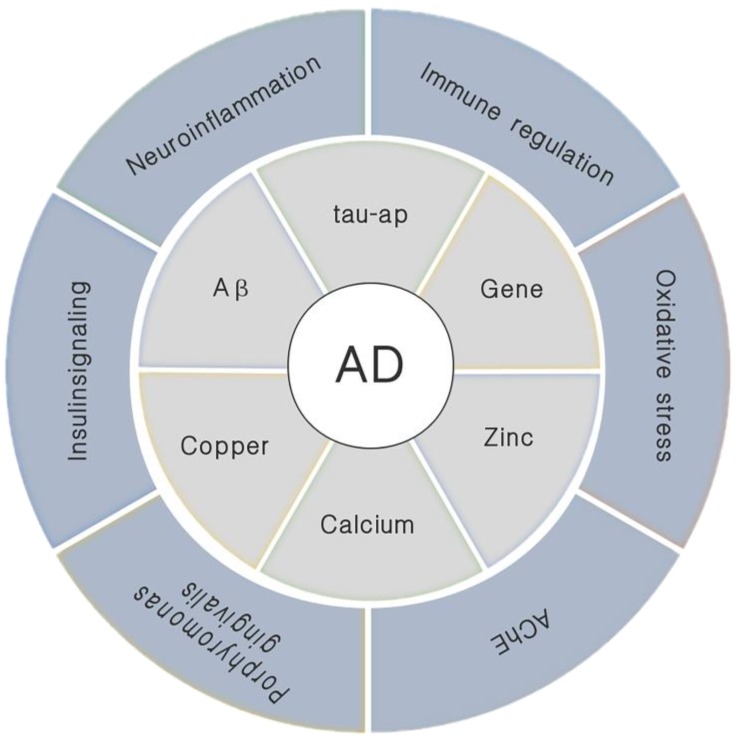
Various pathogenic factors that trigger Alzheimer’s disease (AD).

**Figure 2 molecules-24-01748-f002:**
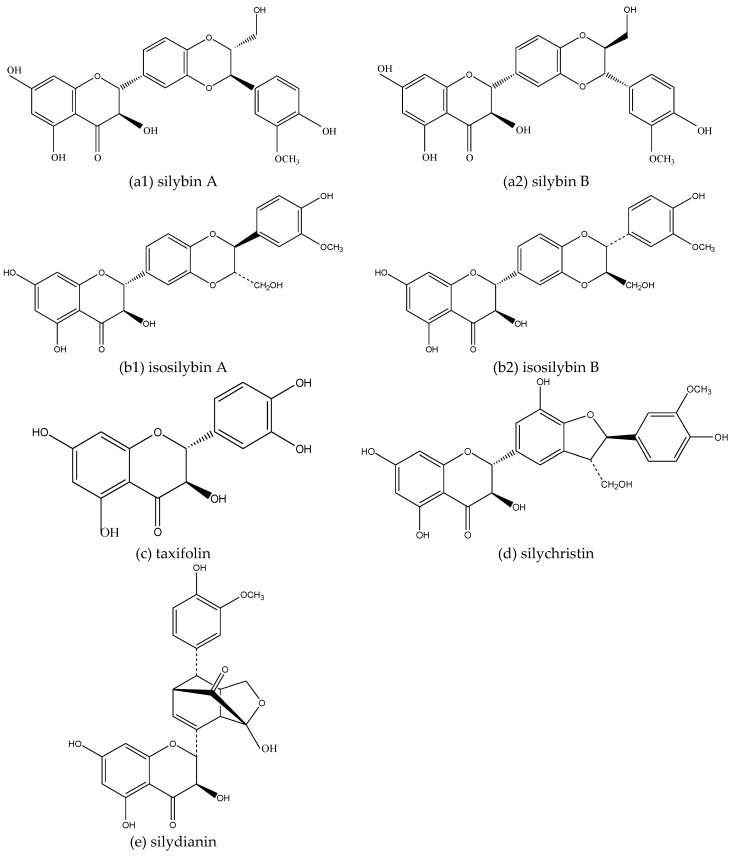
Structure of a monomer component in silymarin: The monomer components contained in silymarin are mainly (**a**) silybin, (**b**) isosilybin, (**c**) taxifolin, (**d**) silychristin, (**e**) silydianin; silybin contains two isomers, (**a1**) silybin A and (**a2**) silybin B respectively; isosilybin also contains two isomers, (**b1**) isosilybin A and (**b2**) isosilybin B respectively.

**Figure 3 molecules-24-01748-f003:**
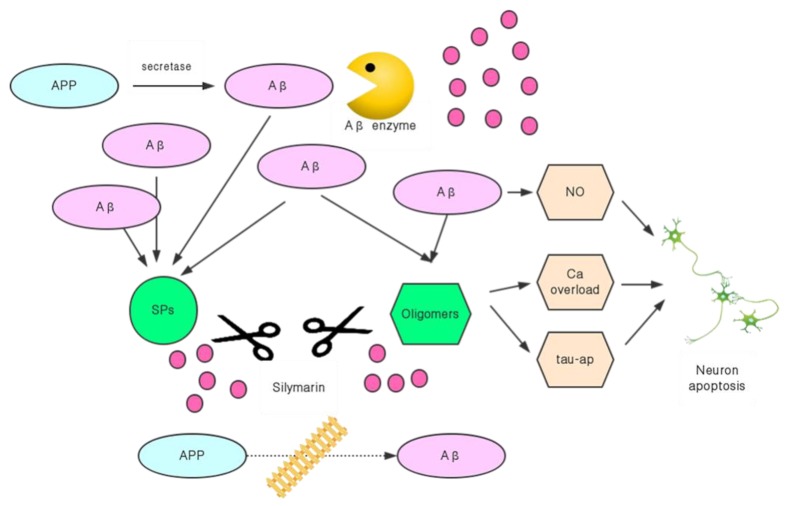
Inhibition of silymarin on Aβ-induced AD process.

**Figure 4 molecules-24-01748-f004:**
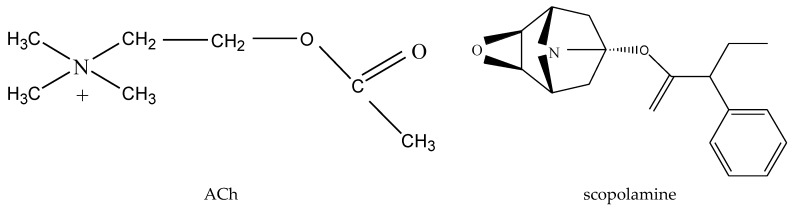
Structure of acetylcholine (Ach) and scopolamine.

**Figure 5 molecules-24-01748-f005:**
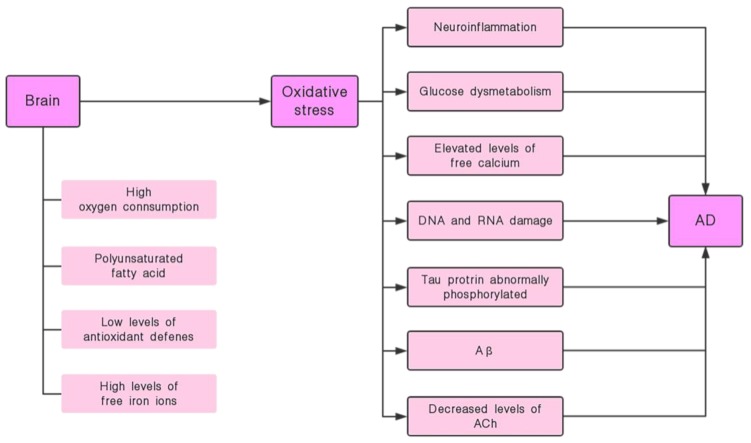
Cerebral oxidative stress leads to impairment of learning and memory.

**Figure 6 molecules-24-01748-f006:**
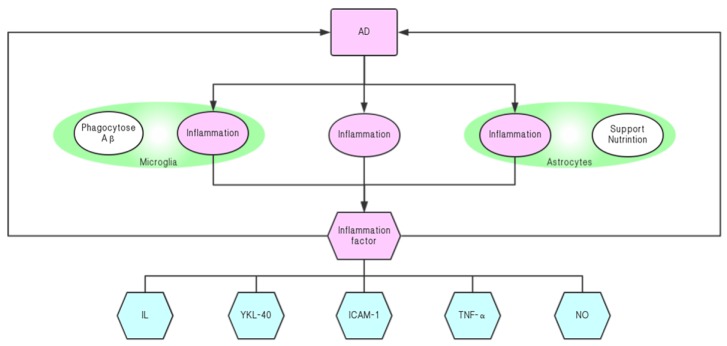
Mutual promotion of neuroinflammation and AD.

**Figure 7 molecules-24-01748-f007:**
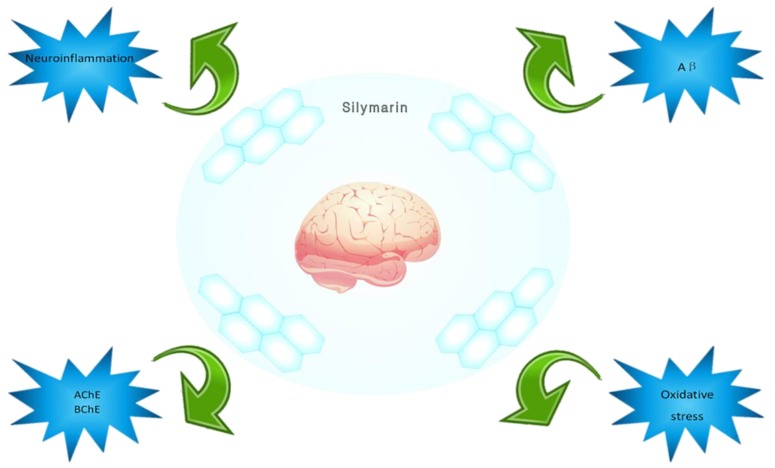
Silymarin’s inhibition and treatment effects for Alzheimer’s disease.

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
