# Peer review of "Silymarin’s Inhibition and Treatment Effects for Alzheimer’s Disease"

_molecules, 2019, doi:10.3390/molecules24091748_

Round 1

Reviewer 1 Report

This short review represents a state-of-art summary of what is known about the neuroprotective effects of Silymarin as a drug candidate for the chemoprevention of AD.

If one excludes some sentences with an awkward English form (see e.g. the first two paragraphs in the introduction), this is a nice review well written/organized that surely deserves publication.

However, a revision by an native English speaking person may be of help. As a minor point I recommend to improve the introduction by discussing some recent issues related to copper-mediated impairment of Aβ proteostasis. See e.g  Coordination Chemistry Reviews 347, 1-22, 2017. This will surely improve a wider coverage of the field.

Author Response

Dear Reviewer

We carefully read and studied your suggestions. We think this is very useful to us and we have modified our article according to these comments. We have marked these changes in the text.

1.          If one excludes some sentences with an awkward English form (see e.g. the first two paragraphs in the introduction), this is a nice review well written/organized that surely deserves publication.

  Thank you very much for your comments. We accepted this comment and made a lot of changes to the preface. These have been marked on the manuscript.

 2.As a minor point I recommend to improve the introduction by discussing some recent issues related to copper-mediated impairment of Aβ proteostasis. See e.g  Coordination Chemistry Reviews 347, 1-22, 2017. This will surely improve a wider coverage of the field.

Thank you very much for your comments, we accepted this comment. We read this article carefully and quoted it in our article.

3. In addition, we scrutinized the article and modified the grammar issues

We are very grateful for your comments and hope to hear your opinion again.

Thank you and best regards.

Yours sincerely,

Hong Guo, Hui Cao, Xiaowei Cui, Wenxiu Zheng, Shanshan Wang, Jiyang Yu and Zhi Chen

Reviewer 2 Report

The authors in this manuscript report the action of Silymarin, a natural product, on Alzheimer disease. The report is interesting, reads well and is logically sound. Some adjustment will improve the quality of the manuscript and capture the attention of a wide audience. In the introduction should be inserted and briefly discuss the so-called amyloid hypothesis and membrane disruption mechanism. For this issue see you this paper doi: 10.1016/j.bbamem.2018.02.022; doi:10.1039/b807980n; 10.1016/j.bpj.2012.06.045; 10.1146/annurev.biochem.75.101304.123901.

In fig.2 the formula structures (a), (b) are redundant, so should be eliminated from the text.

Please use chemical nomenclature according to IUPAC such as at row 89 change peptide with the polypeptide and cut “relative”. Also, row 148 and in the remaining text change polymerization with self-assembly, because polymerization implies the formation of the covalent bond.

Row 176 change condensation with self-assembly.

Row 182-183-184 change fragment with sequence since fragment refers to a polypeptide.

Row 185-188 rephrase this period because in reference 28 was used stereoisomers silybin rather than silymarin and stereochemistry play a crucial role.

Chapter 4 is most significate, so the authors should pay great attention to cite last research such as free phospholipid having short acyl-chains in the aqueous phase (see you doi: 10.1021/acs.jpclett.8b02241; 10.1063/1.4948323; 10.1074/jbc.M116.764092) supporting oxidative stress.

Please change Mn with Mn++ or Mn ions.

Row 355, please add oxidation after arachidonic acid.

Excess of calcium ions plays an important role in the membrane damage as recently reported (see you 10.1039/C8CC01132J).

The authors should be adding in the conclusion section other possible application of silymarin to other diseases in order to do a new way of investigations since oxidative stress is a common factor with type II diabetes and Parkinson as reported in ref 10.1021/acs.jpclett.8b02241. Moreover, type diabetes II and Alzheimer disease are linked as previous reported (see: 10.1002/anie.200904902; 10.1073/pnas.1002555107.

Figure quality should be improved. 

o a polypeptide.

Row 185-188 rephrase this period because in reference 28 was used stereoisomers silybin rather than silymarin and stereochemistry play a crucial role.

Chapter 4 is most significate, so the authors should pay great attention to cite last research such as free phospholipid having short acyl-chains in the aqueous phase (see you doi: 10.1021/acs.jpclett.8b02241; 10.1063/1.4948323; 10.1074/jbc.M116.764092) supporting oxidative stress.

Please change Mn with Mn++ or Mn ions.

Row 355, please add oxidation after arachidonic acid.

Excess of calcium ions plays an important role in the membrane damage as recently reported (see you 10.1039/C8CC01132J).

The authors should be adding in the conclusion section other possible application of silymarin to other diseases in order to do a new way of investigations since oxidative stress is a common factor with type II diabetes and Parkinson as reported in ref 10.1021/acs.jpclett.8b02241. Moreover, type diabetes II and Alzheimer disease are linked as previously reported (see: 10.1002/anie.200904902; 10.1073/pnas.1002555107.

Figure quality should be improved. 

Author Response

Dear Reviewer

We carefully read and studied your suggestions. We think this is very useful to us and we have modified our article according to these comments. We have marked these changes in the text. At the same time, we have made some improvements to the chart section.

1.          In the introduction should be inserted and briefly discuss the so-called amyloid hypothesis and membrane disruption mechanism. For this issue see you this paper doi: 10.1016/j.bbamem.2018.02.022; doi:10.1039/b807980n; 10.1016/j.bpj.2012.06.045; 10.1146/annurev.biochem.75.101304.123901.

  Thank you very much for your comments. We have read these articles carefully and cited them in the article. We briefly discuss the amyloid hypothesis and membrane disruption mechanism in the introduction.

2.          In fig.2 the formula structures (a), (b) are redundant, so should be eliminated from the text.

Thank you very much for your comments, we accepted it and deleted (a) and (b) in Figure 2.

3.          Please use chemical nomenclature according to IUPAC such as at row 89 change peptide with the polypeptide and cut relative. Also, row 148 and in the remaining text change polymerization with self-assembly, because polymerization implies the formation of the covalent bond.

Thank you very much for your comments, we accepted it and made changes in the article, especially row89 and 148. These modifications have been marked in the text

4.          “Row 176 change condensation with self-assembly”, “Row 182-183-184 change fragment with sequence since fragment refers to a polypeptide”, “Row 185-188 rephrase this period because in reference 28 was used stereoisomers silybin rather than silymarin and stereochemistry play a crucial role” and “Row 355, please add oxidation after arachidonic acid”.

Thank you very much for your comments, we accepted them and made changes in the articles according to these comments. Related changes have been marked in the text.

5.          Chapter 4 is most significate, so the authors should pay great attention to cite last research such as free phospholipid having short acyl-chains in the aqueous phase (see you doi: 10.1021/acs.jpclett.8b02241; 10.1063/1.4948323; 10.1074/jbc.M116.764092) supporting oxidative stress.

Thank you very much for your comments, we accepted them and made changes in the articles according to these comments. We have read and studied the literature you provided very seriously and found them very useful. They have enriched our articles and cited them in the article.

6.          Please change Mn with Mn++ or Mn ions.

Thank you for your comments, we have replaced the word "Mn" with "Mn ions"

7.          Excess of calcium ions plays an important role in the membrane damage as recently reported (see you 10.1039/C8CC01132J).

Thank you for your comments, we read this article carefully and found this to be very useful to us. We used it to enrich our article and quote it in the text.

8.          The authors should be adding in the conclusion section other possible application of silymarin to other diseases in order to do a new way of investigations since oxidative stress is a common factor with type II diabetes and Parkinson as reported in ref 10.1021/acs.jpclett.8b02241. Moreover, type diabetes II and Alzheimer disease are linked as previously reported (see: 10.1002/anie.200904902; 10.1073/pnas.1002555107.

Thank you very much for your comments and the literature you provided. We carefully read and cite these documents, and discuss the relationship between AD and PD, type II diabetes, and the possible heroic possibilities of silymarin in these two diseases.

9.          Figure quality should be improved.

We accepted your comments and made changes to the figure. Especially Figure 1.

  If you vent other questions, please point out to us, we will be grateful.

We are very grateful for your comments and hope to hear your opinion again.

Thank you and best regards.

Yours sincerely,

Hong Guo, Hui Cao, Xiaowei Cui, Wenxiu Zheng, Shanshan Wang, Jiyang Yu and Zhi Chen

Reviewer 3 Report

Present study presents an overview of scientific papers dealing with the mechanisms of the onset of Alzheimer disease and pathological consequences in brains of humans. In individual parts they deal with pathogenic factors that can trigger diseases.  Besides, authors summarise experimental and clinical work published in scientific journals on the effects of silymarin on key markers and processes of this disease. Present review is original contribution to the mechanism of action of SIL on pathogenesis of Alzheimer´s disease (AD) .

The structure of MS is appropriately chosen and the individual parts are written in clear, understandable style. To my opinion some paragraphs with background information about pathogenesis of AD could be more concise, for example lines 206-267 or 391-430. There is repetition of some information.

Graphical part is well designed.

However the main drawback of MS is English grammar, especially the composition of sentences (syntax).  The whole text of MS needs careful editing by professionals or native English speaking scientists.

some expression sounds strange, for example :

Line 188: do not use word “ingredients” regarding individual components of silymarin

Line 374: “acute administration” is not suitable name,

Silymarin should be written in lowercase letter through whole text

Author Response

Dear Reviewer

  We carefully read and studied your suggestions. We think this is very useful to us and we have modified our article according to these comments. We have marked these changes in the text.

  1. We refer to your comments for the pathogenesis of AD (lines 206-267 or 391-430 in the previous version of the article) and have modified your other questions. At the same time, in the process, we met the opinions of other experts and suggested that we add some information. We got the current version of the article after making the changes. We would be grateful if you could get your valuable feedback again.

2. “Line 188: do not use word ingredients regarding individual components of silymarin”

and “Line 374: acute administration is not suitable name”

  Thank you very much for your comments, we accepted it and modified the article according to this opinion.

3. Silymarin should be written in lowercase letter through whole text.

Thank you very much for your comments. We checked the article and changed the silymarin in the text to lowercase.

4. In addition, we scrutinized the article and modified the grammar issues

We are very grateful for your comments and hope to hear your opinion again.

Thank you and best regards.

Yours sincerely,

Hong Guo, Hui Cao, Xiaowei Cui, Wenxiu Zheng, Shanshan Wang, Jiyang Yu and Zhi Chen

Round 2

Reviewer 3 Report

Authors accepted suggestions of reviewers and many parts have been re-written by adding new information. After detailed description of pathogenesis of AD, the multitarget effects of silymarin or silybin are reported. MS is interesting and suitable for publication. However most of text does not seem to be edited by professional editing service.

Author Response

Authors accepted suggestions of reviewers and many parts have been re-written by adding new information. After detailed description of pathogenesis of AD, the multitarget effects of silymarin or silybin are reported. MS is interesting and suitable for publication. However most of text does not seem to be edited by professional editing service.

Thank you very much for your comments, we accepted it and modified the article according to this opinion. We applied professional English editing service, and asked the professionals to make changes to the grammar of this article. These modification points are marked
